# Evaluation of the Potential Allergenicity of Strawberries in Response to Different Farming Practices

**DOI:** 10.3390/metabo10030102

**Published:** 2020-03-12

**Authors:** Mateusz Aninowski, Renata Kazimierczak, Ewelina Hallmann, Joanna Rachtan-Janicka, Elżbieta Fijoł-Adach, Beata Feledyn-Szewczyk, Iwona Majak, Joanna Leszczyńska

**Affiliations:** 1Institute of Institute of Natural Resources and Cosmetics, Faculty of Biotechnology and Food Sciences, Lodz University of Technology, Stefanowskiego 4/10, 90-924 Lodz, Poland; mateusz.aninowski@edu.p.lodz.pl (M.A.); joanna.leszczynska@p.lodz.pl (J.L.); 2Institute of Human Nutrition Sciences, Department of Functional and Organic Food, Warsaw University of Life Sciences, Nowoursynowska 159c, 02-776 Warsaw, Poland; renata_kazimierczak@sggw.pl (R.K.); joanna_rachtan_janicka@sggw.pl (J.R.-J.); 3Institute of Soil Science and Plant Cultivation-State Research Institute, Department of Systems and Economics of Crop Production, Czartoryskich 8, 24-100 Pulawy, Poland; elaf.bea@poczta.onet.pl (E.F.-A.); beata.szewczyk@iung.pulawy.pl (B.F.-S.); 4Institute of Food Technology and Analysis, Lodz University of Technology, Stefanowskiego 4/10, 90-924 Lodz, Poland; iwona.majak@p.lodz.pl

**Keywords:** potential allergenicity, anthocyanins, Bet v1, conventional, integrated, organic, profilins, strawberry

## Abstract

Fruit allergies are a large problem today. Many consumers suffer from food allergies or intolerances. The method of food production has a major impact on its quality. In fruit and vegetable production, three different farming systems are mainly found: conventional, integrated pest management and organic production. The latter is considered the best in terms of fruits and vegetables safety and high quality. The present experiment was performed to demonstrate the effect of the strawberry production method on its allergenic properties and flavonoid content. The strawberry ‘Honeoye’ cv. was used for the study. Fruits from the three cultivation systems, organic, conventional and integrated, were tested for their content of biologically active compounds and their potential allergenicity. The results obtained indicate that the strawberries from the organic system were the safest because they contained the lowest levels of Bet v1 and profilin in comparison with the fruits from the integrated and conventional systems.

## 1. Introduction

Strawberries are one of the most popular berry fruits in the world. Many studies have shown that regular fruit consumption can diminish the risk of many chronic diseases, such as hypertension, different kinds of cancer, cardiovascular diseases and overweight and obesity [1,2,3,4] due to their bioactive compound content, especially phenolic compounds and compounds with antioxidant properties [5,6,7]. According to the organic practices rules and European law, in organic farming systems, the use of pesticides and synthetic fertilizers are completely forbidden [8]. Only natural fertilization methods are used (compost, green manure as well animal manure). For pest management, plants produce “natural pesticides” better known as phenolic compounds [9,10]. Integrated pest management (IPM) involves several treatments against pests and diseases. Mineral fertilization is allowed along with animal and green manure. For plant protection pesticides are allowed, but in limited quantity and quality [11]. Organically produced strawberries contain more bioactive compounds compared to conventional strawberries [12]. Integrated strawberry fruits are characterized by the lowest levels of phenolic compounds and antioxidants compared to conventional and organic fruits [13]. Strawberries are recognized by the consumer as very tasty and healthy fruits. Many studies have shown that organic farming systems can create high-quality fruits [14]. There is little known about the effects of integrated farming systems on strawberry fruit quality. However, it should be remembered that the consumption of strawberries brings the risk of food allergies [15]. During the last decades, the prevalence of allergies has increased drastically, whereby 3–4% of the adult population and up to 6% of children are affected by food allergies [16]. In Europe, sensitivity to Bet v1, the major allergen from birch (*Betula verrucosa*) pollen, dominates. Most consumers with an allergy to strawberries could suffer allergies to other berry fruits. In contrast, some allergic reactions to strawberries are not typical allergies but represent a kind of food intolerance [17]. Therefore, the best form of therapy to avoid the symptoms of strawberry allergies is to exclude them from the diet [18,19]. Some studies have shown that the levels of allergens found in strawberries are too low to justify their exclusion from the diet [20]. Some authors consider these foods, along with others such as milk, shellfish, and eggs, as endogenous histamine releasers, although the mechanisms are still not well understood [21].

Currently, seven allergenic proteins, including different isoforms, have been found in strawberry fruits. The main allergen in strawberry (*Fragaria × ananassa*) belongs to the PR-10 group (17  kDa) and is homologous to the major birch pollen Bet v 1. Based on the latest knowledge, several isoforms of this protein have been described in strawberries [22]. They may be responsible for allergic reactions in sensitive individuals as well as reactions caused by the cross-reactivity of allergens. The second group of non-specific lipid transfer proteins includes the profilin. They are mainly responsible for strawberry allergies occurring in the Mediterranean area [23,24]. The profilins family contains conserved protein sequences with molecular masses ranging from 12 to 15 kDa, corresponding to polypeptides with 124−153 amino acids [25].

In the present study, we want to find relationship between the higher anthocyanin concentration and potential allergenicity in “Honeoye” cv. strawberries from organic, integrated and conventional production. There is a complete lack of information about the bioactive compound content and potential allergenicity of strawberries in the literature. The main hypothesis of the work was to find a link between the flavonoid content and the level of homologue allergens (Bet v1) in strawberry fruits. The aim of this study was to determine whether the use of three types of cultivation (organic, integrated and conventional) has an impact on the quality and potential allergenicity of strawberries.

## 2. Materials and Methods 

### 2.1. Fruits Origins

To carry out the present experiment to show the impacts of three cultivation systems on the flavonoid content and allergy potency of strawberries, one cultivar of strawberry, ‘Honeoye’ cv, was used. Plants were cultivated using three systems, organic, integrated and conventional, in farms located in the Lublin region in 2013 (Table 1). Four farms were chosen for each cultivation system. Two independent, 1 kg samples were used from each farm. All rules of organic, integrated and conventional practices were used in the selected farms. Strawberry fruits were collected in their fully ripened stage during 28–31 May 2013 from all experimental farms. The data about the weather conditions in the experimental farms are presented in Figure 1. 

### 2.2. Plant Material Preparation

The fresh fruits from all experimental farms were transported to the chemical laboratory for analysis purposes. The fruits were gently washed, and the green stalk was cut off. Then, each fruit was cut into two parts. One-half was used for dry matter determination, and one was used for freeze-drying using a LabconCo (2.5 L, Warsaw, Poland) freeze-drier (conditions 0.110 mBa, −50 °C). After freeze-drying, the plant material was ground in a laboratory mill (A-11, IKA^®^, Retsch, Germany). The ground samples were placed into plastic tubes and kept at −80 °C.

### 2.3. Dry Matter Content

The fresh strawberries were cut into small cubes. Next, 3 g of tissue was placed into a glass tube under the following drying conditions: 72 h, 105 °C in a dryer (Farma Play FP-25W, Warsaw, Poland). After 3 days, the samples were cooled in a desiccator with dry silica gel and weighed. The actual mass of the samples was noted. Next, the drying process was applied again for 24 h. The re-drying process was repeated two more times to achieve a stable and constant weight. The dry matter content of the strawberries was calculated in grams per 100 g of fresh weight (FW) as described by Polish Norm PN-EN 12145:2001 [26].

### 2.4. Flavonoids Content

Polyphenols were measured by a high performance liquid chromatography (HPLC) method [27]. Powdered dry strawberry samples (50 mg) were extracted with 80% methanol in an ultrasonic bath (35 °C, 15 min, 5500 Hz). Next, the samples were centrifuged (2 °C, 10 min, 6000 rpm). The supernatant was then collected and re-centrifuged (−1 °C, 10 min, 14,000 rpm). One millilitre of supernatant was transferred to HPLC vials and used for analysis. For analysis purposes, the following HPLC set-up was used: two LC-20AD pumps, a CMB-20A system controller, an SIL-20AC autosampler, an ultraviolet-visible SPD-20AV detector, a CTD-20AC oven, and a Phenomenex Fusion-RP 80A column (250 × 4.60 mm); all of the components were from Shimadzu (Shimpol, Poland). The gradient mobile phase contained 10% (phase A) and 55% (phase B) acetonitrile with ultrapure-water. After obtaining a stable value (3.0) with orthophosphoric acid, the phases were ready to flow: 1 mL min−1, time programme: 1.00–22.99 min 95% phase A and 5% phase B, 23.00–27.99 min 50% phase A and 50% phase B, 28.00- 28.99 min 80% phase A and 20% phase B, and 29.00–38.00 min 95% phase A and 5% phase B. The wavelengths used for detection were 250 nm for flavonoids (quercetin-3-O-rutinoside, kaempferol-3-O-glucoside, myricetin, quercetin, quercetin-3-O-glucoside, luteolin, kaempferol) and 370 nm for phenolic acids (gallic, chlorogenic, caffeic, *p*-coumaric, ferulic). All phenolic compounds were identified based on Fluka (Warsaw, Poland) and Sigma Aldrich (Warsaw, Poland) external standards with a purity of 99.5% (Figure 2).

### 2.5. Anthocyanins Content

The first step of the anthocyanin extraction (in 80% ethanol) was as described in the case of the polyphenols (flavonoids). Next, 2.5 mL of supernatant was put into new plastic tubes and 2.5 mL of 10 M HCl and 5.0 mL 100% methanol were added. Next, the samples were kept in the refrigerator for 5 min. One millilitre of extract was used for HPLC analysis. The elution of anthocyanins was performed using isocratic flow with one mobile phase: 5% acetic acid, acetonitrile, and methanol (70:10:20). The flow rate was 1.5 mL min^−1^, and the wavelength range for detection was 530 nm. Anthocyanins were identified based on Sigma-Aldrich external standards (pelargonidin-3-O-rutinoside, pelargonidin-3-O-glucoside, cyaniding-3-O-glucoside) with a purity of 99.9% (Figure 3) [28].

### 2.6. Potential Allergenicity Analysis

The following reagents were used: mouse antibodies against Bet v1 (Dendritics, Lyon, France), rabbit antibodies against plants profilin (Dendritics), a conjugate of antibodies against mouse immunoglobulins with alkaline phosphatase (Sigma-Aldrich), and antibodies against the rabbit immunoglobulin conjugate with alkaline phosphatase (Sigma Aldrich). Additionally, a 3% solution of commercial skim milk (Piątnica, Warsaw, Poland) in deionized H_2_O, pNPP (Sigma-Aldrich) was used as the substrate for the alkaline phosphatase, with 3 M NaOH (Sigma-Aldrich) as the stop reagent and PBS with 0.1% Tween 20 (Sigma-Aldrich) as a washing agent. In presented manuscript only determination of potentially allergenic protein homologous to Fra a 1 by means of cross-reaction with antibodies against Bet v 1 have been done.

The analyses were performed according to previously described methodologies [27]. Briefly, the Total Protein Extraction Kit for Plant Tissues was used to obtain the fruit protein extracts. Then, the potential allergen content was determined by indirect, non-competitive ELISA. First, the plate (SPL Lifesciences, Seul, Korea) was coated with the standard solution or the extract samples. Each well of the microplate was filled with 100 μL of the standard solution or the extract samples, and the plate was kept in a refrigerator (4 °C) for 12 h. After washing with PBS buffer, 400 μL of 3% skim milk in PBS solution was added to the wells, and the plate was then incubated for two hours. Then, after washing, 100 μL of the antibodies against Bet v1 or profilin was added and incubated for one hour at room temperature. After one hour, the plate was washed with the washing buffer. Then, 100 μL of the anti-mouse antibody for Bet v 1 homologue determination (or anti-rabbit in the case of profilin determination) that was conjugated to alkaline phosphatase was added to each well and incubated for one hour. Finally, the plate was washed with the washing buffer, and 100 μL of the substrate pNPP for enzymes was added. After 30 min, the reaction was stopped by adding 100 μL of the stopping solution. The absorbance was read at 405 nm with the use of a Multiscan RC microplate reader (Labsystems, Helsinki, Finland). Then, the results were calculated using a standard curve prepared with either the Bet v 1 allergen or profilin.

### 2.7. Statistical Analysis

The obtained results are expressed as the mean values for the organic, integrated and conventional systems for the ‘Honeoye’ strawberry cultivar. The mean values for the organic, integrated and conventional strawberries were each obtained from 8 individual measurements (four farms, two independent samples from each field). The statistical calculations were based on a one-way analysis of variance with the use of Tukey’s test (*p *= 0.05) using Statgraphics 5.1 software (StatPoint Technologies, Inc., Warranton, VA, USA). A lack of statistically significant differences between the examined groups is indicated by the use of the same letters. A standard error (SE) was given with each mean value. To obtain a clearer picture of the interrelations between the identified biologically active compounds and the allergenic proteins, an analysis of the main components was used. A principal component analysis (PCA) is a useful statistical tool that applies an orthogonal transformation to convert a set of data of possibly correlated variables into a set of values in linearly uncorrelated variables called principal components. The PCA figures were made using Statistica 10.0 (Statsoft, Tulsa, OK, USA). The PCA was conducted using the correlation matrix, not the covariance matrix, which corresponds to the analysis of standardized data. 

## 3. Results

The presented experiment provides very important information about the impacts of growing systems on the quality of strawberry fruits. The authors tried to use a holistic and multi-directional “from farm to fork” approach. The obtained results showed not only the content of biologically active compounds in strawberry fruits but also that the type of strawberry production can affect consumers who suffer from allergy problems. The result of the Principal Component Analysis (PCA) analysis for the strawberries produced in the three different cultivation systems showed a high overall variation rate explained by PC1 and PC2, which was 54.96% (Figure 2). 

The degree of dependence between the ‘Honeoye’ strawberries from the integrated production system and the factors marked as (TA) total anthocyanins, (Cy-3O-G) cyanidine-3-O-glucoside, (Pel-3O-G) pelargonidine-3-O-glucoside, (Bet v1) Bet v1 and (profilin) profilin are particularly important. As shown in the graph, integrated, organically and conventionally produced strawberries were located in completely separate parts of the PCA plot. This suggests significant dissimilarity in the quantitative composition of biologically active compounds identified and marked in the strawberries. No significant effects of the cultivation system on the dry matter and total polyphenol contents in ‘Honeoye’ strawberries were found. However, it was observed that fruits obtained from the integrated system were characterized by a significantly higher content of phenolic acids (*p *< 0.05) (Table 2). 

Chlorogenic acid was dominant in the pool of identified polyphenolic acids. Integrated fruits were characterized by a significantly higher content of chlorogenic acid in comparison with conventional and organic fruits, which were 100.12 mg 100 g^−1^ DW, 96.50 mg 100 g^−1^ DW and 72.03 mg 100 g^−1^ DW respectively (Table 2). The gallic acid contents of the strawberry fruits were low, but significant differences were observed between the studied cultivation systems. Strawberries from conventional (*p *< 0.05) cultivation contained significantly more gallic acid compared with integrated (*p *< 0.05) and organic fruits (*p* < 0.05). There were no significant differences in the ferulic acid content in strawberry fruits from the three different cultivation systems. However, there was significantly more ellagic acid in the fruits of the integrated strawberries (28.10 mg 100 g^−1^ DW) compared with the organic (21.12 mg 100 g^−1^ DW) but not conventional (27.03 mg 100 g^−1^ DW) fruits. We did not observe statistical difference in the total flavonoids content between examined strawberries from different cultivation systems. Seven compounds from the flavonols and three from anthocyanins groups were identified in the strawberry fruits (Figure 3 and Figure 4). 

Very high variation was found in the quercetin-3-O-glucoside contents. The quercetin-3-O-glucoside content in the conventional fruits was 90.92 mg 100 g^−1^ DW, while it was as much as 10 times lower in integrated fruits (9.09 mg 100 g^−1^ DW), and it was 29.69 mg 100 g^−1^ DW in organic fruits. Organic fruits contained significantly more kaempferol-3-O-glucoside compared to fruits from the two other experimental combinations (*p *< 0.05). Fruits from the conventional system were characterized by a significantly higher myrycetin (*p* < 0.05) and luteolin (*p* < 0.05) contents in comparison with fruits from the organic and integrated systems. However, the organic fruits had the highest contents of kaempferol and quercetin (Table 2).

Anthocyanins belong to the group of flavonoid compounds. Significantly more total anthocyanins (*p* < 0.05) were found in integrated strawberry fruits (199.27 mg 100 g^−1^ DW) compared with organic fruits (153.38 mg 100 g^-1^ DW) and conventional fruits (155.51 mg 100 g^−1^ DW). The main anthocyanin pigment identified in strawberry fruits was cyanidin-3-O-glucoside. The fruits from the integrated production were characterized by a significantly higher (*p* < 0.05) content of this compound (163.27 mg 100 g^−1^ DW) in comparison with conventional (120.96 mg 100 g^−1^ DW) and organic (119.52 mg 100 g^−1^ DW) fruits. The second most important pigment found in the fruits was pelargonidin-3-O-glucoside. The highest and significant concentration of that pigment was observed in integrated strawberry (*p *< 0.05). The content of Bet v1 was significantly dependent on the growing system (*p* < 0.05). Significantly less Bet v1 was found in organically produced fruits, while integrated and conventional fruits were characterized by similar and much higher contents, which were 678.28 ng g^−1^ and 673.29 ng g^−1^, respectively. There were no significant differences in the profilin contents of the strawberry fruits from the three cultivation systems. However, it is clearly visible that the integrated fruits were characterized by the highest content of these homologues of allergenic proteins (Table 2).

## 4. Discussion

The strawberry is one of the most popular fruits in the world. Consumers recognize them as one of the tastiest and healthiest fruits. Strawberries are one of the best sources of bioactive compounds, such as polyphenols (especially flavonols and anthocyanins) and vitamin C [29,30,31]. Organic agriculture is considered one of the best alternatives for sustainable and good quality food production. The first and most important rule forbids the use of chemical fertilizers and artificial pesticides. Integrated pest management, better known as IPM in fruit production, is a trend to reduce the levels of agricultural chemistries to protect consumer health and the natural environment [32,33]. Many experiments conducted to compare organic and conventional farming practices have shown that organic fruits are richer in biologically active compounds [13], [34,35]. However, the quality of fruits from IPM (integrated pest management) systems has not been assessed yet, even though IPM was introduced for fruit cultivation long ago [36,37]. Our results are partially in accordance with previous studies for the total flavonol contents as well for individual flavonol compounds such as quercetin-3-O-rutinoside, kaempferol-3-O-glucoside and aglycone forms (Table 2). The fact that organically produced fruits contain more bioactive compounds can be explained by the C/N theory [38]. Plants cultivated in organic systems, where nitrogen is a limiting growth factor immediately start to produce non-N-containing secondary metabolites as polyphenols. Plants in organic farm systems grow much more slowly, but their fruits contain more bioactive compounds compared to fruits from conventional plant production systems. On the other hand, the high levels of polyphenols in plant tissues is an effect of biotic stress factors such as pests, diseases or increased UV-radiation [39]. Plants produce more polyphenolic compounds, also known as “natural pesticides”, and antioxidants to protect their foliar photosynthetic systems against damage [40]. Many factors influence the content of polyphenolic compounds in strawberry fruits. In some cases, the obtained results cannot confirm the previously described theory (C/N). Conventional strawberries were characterized by lower level of total polyphenols compared to the sustainable systems (integrated), and it were 152 mg 100 g^−1^ DW and 165 mg 100 g^−1^ DW, respectively [34]. In the present experiment, the same effect was obtained (Table 2). Conventional strawberries contained 503.49 mg 100 g^−1^ DW, integrated 499.76 mg 100 g^−1^ DW and organic 478.11 mg 100 g^−1^ DW. Similar results were also obtained for organically and integrated produced strawberries, which contained fewer total polyphenols [11]. The level of total polyphenols is very variable and dependent on genetic factors. Four new strawberry cultivars represent a total polyphenol content in the range of 989.65 mg 100 g^−1^ DW for Zamorana cv. to 481.26 mg 100 g^−1^ DW for Pacal cv. [41]. In the present experiment, we obtained levels similar to those of the new cultivars, which were lower but still satisfied the previously described theory (Table 2). The phenolic acids in the strawberry fruits were represented by gallic, chlorogenic, *p*-coumaric, ferulic and ellagic acids. Chlorogenic acid is the predominant phenolic acid [42,43]. The chlorogenic acid content was strongly dependent on the cultivation system. Integrated strawberries contained significantly more chlorogenic acid (100.12 mg 100 g^−1^ DW) compared with organic (72.03 mg 100 g^−1^ DW) and conventional (96.50 mg 100 g^−1^ DW) strawberries (Table 2). The cultivation system can significantly influence the chlorogenic acid content in the fruits. Organically produced tomatoes contained less chlorogenic acid (1.45 mg 100 g^−1^ DW) compared to conventional tomatoes (2.26 mg 100 g^−1^ DW) [44].

In strawberry fruits, anthocyanins play a very important role. They are the most numerous group among the flavonoid compounds. Many studies have shown that high anthocyanin levels in strawberry fruits are an effect of plant response to UV radiation [45,46]. The anthocyanin synthesis pathway is regulated primarily at the transcriptional level and is induced by UV light. In contrast, anthocyanins that provide plants with UV tolerance are regulated by distinct special genes that are strongly regulated by UV light. It should be noted that exposure of plants to sunshine and the number of sunshine hours during the day play a significant role in the production of anthocyanins in strawberry fruits. In the present experiment, integrated strawberries contained significantly more total anthocyanins (199.29 mg 100 g^−1^ DW) compared to organic (153.38 mg 100 g^−1^ DW) and conventional (155.51 mg 100 g^−1^ W) strawberries. However, the farms from which the fruits were harvested for research were located next to each other, in a similar region (Table 1). According to the collected data on the weather conditions, it is clearly visible that there was much more sun exposure in the integrated farms (May and June) compared to the organic farms. In another experiment with the reaction of strawberries to UV radiation, fruits with higher exposure to sunlight produced much more pelargonidin-3-O-glucoside, pelargonidin-3-rutosinide, and cyanidin-3-glucoside [47]. We found the same effect (Table 2). In another presented investigation, strawberries form organic production were found to have more total anthocyanins (124.63 mg 100 g^−1^ DW) compared to integrated strawberries (68.44 mg 100 g^−1^ DW) [41]. We observed some relationship between the Bet v1 content and the total anthocyanins, especially in integrated strawberry fruits (Figure 5B). In addition, the second allergy homologue (profilins) was connected to the total anthocyanin content (Figure 5E). The weak effect between Bet v1 and profilins and the content of total anthocyanins in conventional strawberries was not significant (Figure 5C and Figure 5D). The obtained results are a very preliminary observation and further research will be needed to understand the biological processes of this phenomenon. Anthocyanin compounds in strawberry fruits are responsible for causing allergic reactions. It was observed that allergic reactions occurred at low levels when red strawberries were compared to a colourless (white) strawberry mutant. White strawberries, known to be tolerated by individuals affected by allergy, were found to be virtually free of the strawberry allergen. Additionally, several enzymes in the pathway for flavonoid biosynthesis were differentially expressed and down regulated compared with purple anthocyanin colour [48,49]. In such situations, the lower levels of pigment are safer when compared to highly pigmented fruits. This information is a very important f the consumer suffer rather on food intolerance than allergy. We found in organically produced strawberries with the lowest amounts of anthocyanins and Bet v1 levels, but not profilins (Table 2). In addition to natural allergens, the plants produced in conventional systems may contain many residual chemicals (e.g., pesticides or derivatives), that also toxic and allergenic properties. The IPM production system limits the use of certain pesticides, food additives and preservatives, but they are still used and have a negative impact on human health [50,51]. Among the chemicals that have a negative impact on human health, pesticides are the most serious. Pesticides may have carcinogenic, neurodegenerative and allergic effects [52,53]. Plants produced under organic conditions with very strict rules regarding the use of artificial, harmful substances, genetic modification and all other forbidden substances and techniques, are safer and less allergenic compared to conventional or integrated plants. Many consumers believe that organic fruits and vegetables are safer for their health. The present results are in accordance with their confidence, especially in the case of strawberry fruits. We found that strawberries form organic farming system had the lowest levels of the allergen Bet v1 and the protein homologue profilin compared to conventional and integrated strawberries. There is limited experimental evidence showing such problems in the available literature. Most of the available studies showing the differences between organic and conventional production systems are mainly concerned with fruit quality and yield [11,13,34,52]. Only two previous investigations addressed the problem of allergies in organic and conventional crops, but not in IPM. Organically produced apricots had a greater allergy potency. This was reflected by the higher levels of Bet v1 and profilins in organic fruits compared to conventional fruits due to the higher polyphenol content [27]. On the other hand, organically cultivated tomatoes were characterized by higher profilin content but not Bet v1 content [53]. Contrary results were obtained in the present study. Organically produced strawberries were found with the lowest Bet v1 levels (376.4 ng g^−1^) compared to integrated (678.28 ng g^−1^) and conventional strawberries (637.29 ng g^−1^). IPM-produced strawberries are characterized by the highest levels of Bet v1 and profilins, and as we noted, they are then connected with the highest level of anthocyanins. Fully ripened red tomatoes cause a stronger allergic reaction compared to orange ripened tomatoes (not fully red). It seems that the allergic reaction could be connected with carotenoid pigments such as lycopene. In cases of tomatoes from organic farming system, the skin test confirmed the higher levels of potential allergenicity of the vegetables. Using genetic and breeding approaches, it is possible to search the numerous Fragaria genotypes for allergen proteins. It turns out that some wild varieties appeared devoid of the allergen. Thus, breeders can take advantage of the biodiversity of Fragaria to select for hypoallergenic strawberry lines [54]. On the other hand, based on the results obtained, we can take advantage of the fact that organically produced strawberries contained a lower level of Bet v1. Considering other health-related aspects of strawberries from organic farming, this second solution is much more beneficial.

## 5. Conclusions

In summary, our experiment support the hypothesis that organically produced strawberry fruits are safer because they are less allergenic than conventional and integrated fruits. On the other hand, the second theory that organic plant cultivation increases the level of bioactive compounds was only partially confirmed. IPM plant production systems even excluded some chemical agents used in plant protection systems that are still similar to conventional systems. The results suggest that it is more valuable to use organically produced strawberries because of their safety and health properties.

## Figures and Tables

**Figure 1 metabolites-10-00102-f001:**
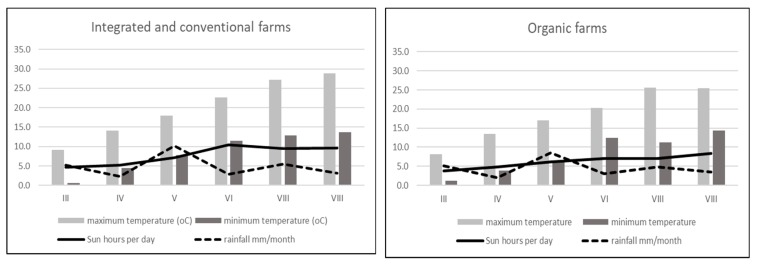
Weather conditions in experimental farms (organic, integrated and conventional) 2013 in time of strawberry fruits development.

**Figure 2 metabolites-10-00102-f002:**
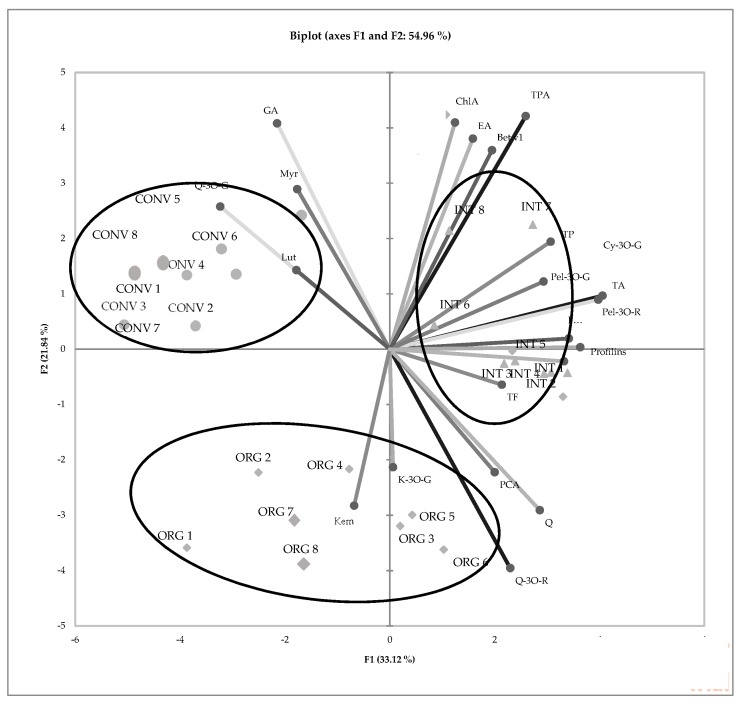
PCA analysis showing the relationship between the chemical composition and allergy potency of ‘Honeoye’ strawberry from three cultivation system. (TP) total polyphenols, (TPA) total phenolic acids, (GA) gallic acid, (ChlA) chlorogenic acid, (pCA) *p*-coumaric acid, (FA) ferulic, (EA) ellagic acid, (TF) total flavonoids, (Q-3O-R) quercetin-3-O-rutinoside, (Q-3O-G) quercetin-3-O-glucoside, (K-3O-G) kaempferol-3-O-glucoside, (Myr) myrycetin, (Lut) luteolin, (Q) quercetin, (Kem) kaempferol, (TA) total anthocyanins, (Cy-3O-G) cyanidin-3-O-glucoside, (Pel-3O-R) pelargonidin-3-O-rutinoside, (Pel-3O-G) pelargonidyno-3-glucoside, (Bet v1) Bet v1, (profilins) profilins; ORG 1–8 (organic management); INT 1–8 (integrated farm management); CONV 1–8 (conventional farm management)

**Figure 3 metabolites-10-00102-f003:**
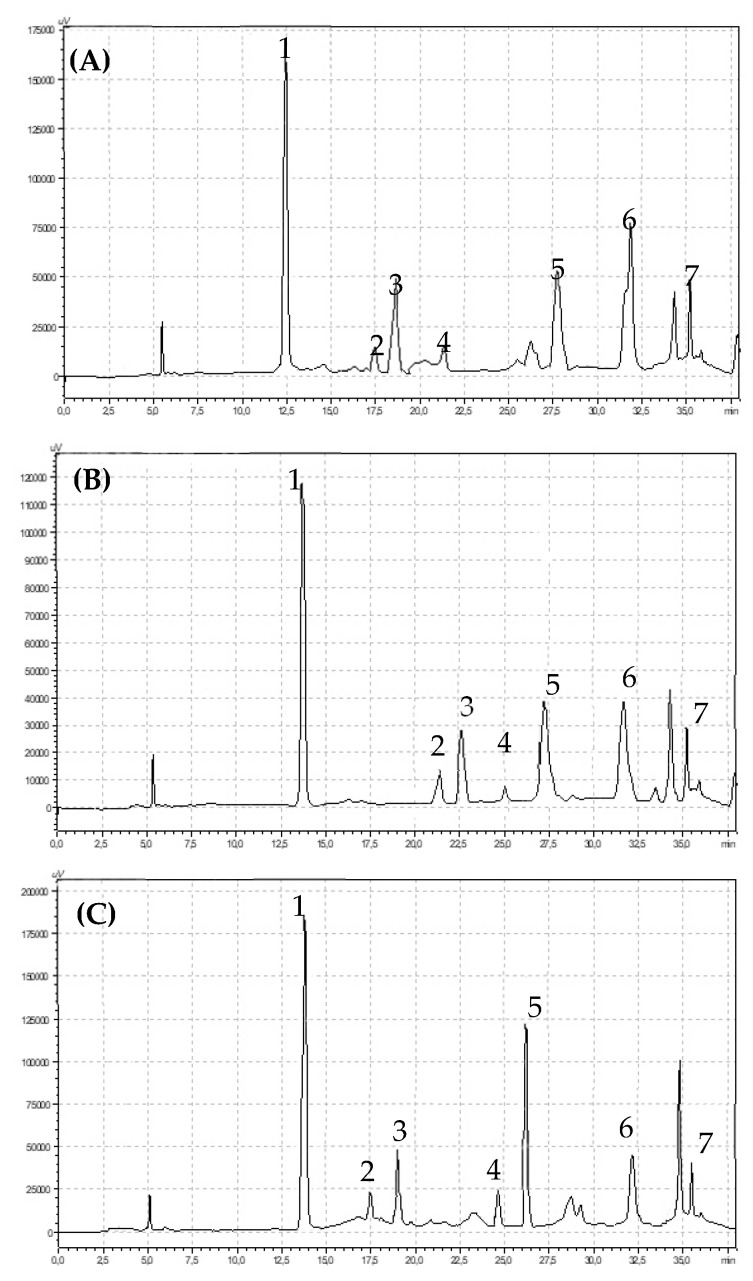
Chromatograms showing retention times for flavonols identified in (**A**) organic (**B**) integrated and (**C**) conventional strawberries (1) quercetin-3-O-rutinoside, (2) myrycetin, (3) quercetin, (4) luteolin, (5) quercetin-3-O-glucoside, (6) kaempferol-3-O-glucoside, (7) kaempferol.

**Figure 4 metabolites-10-00102-f004:**
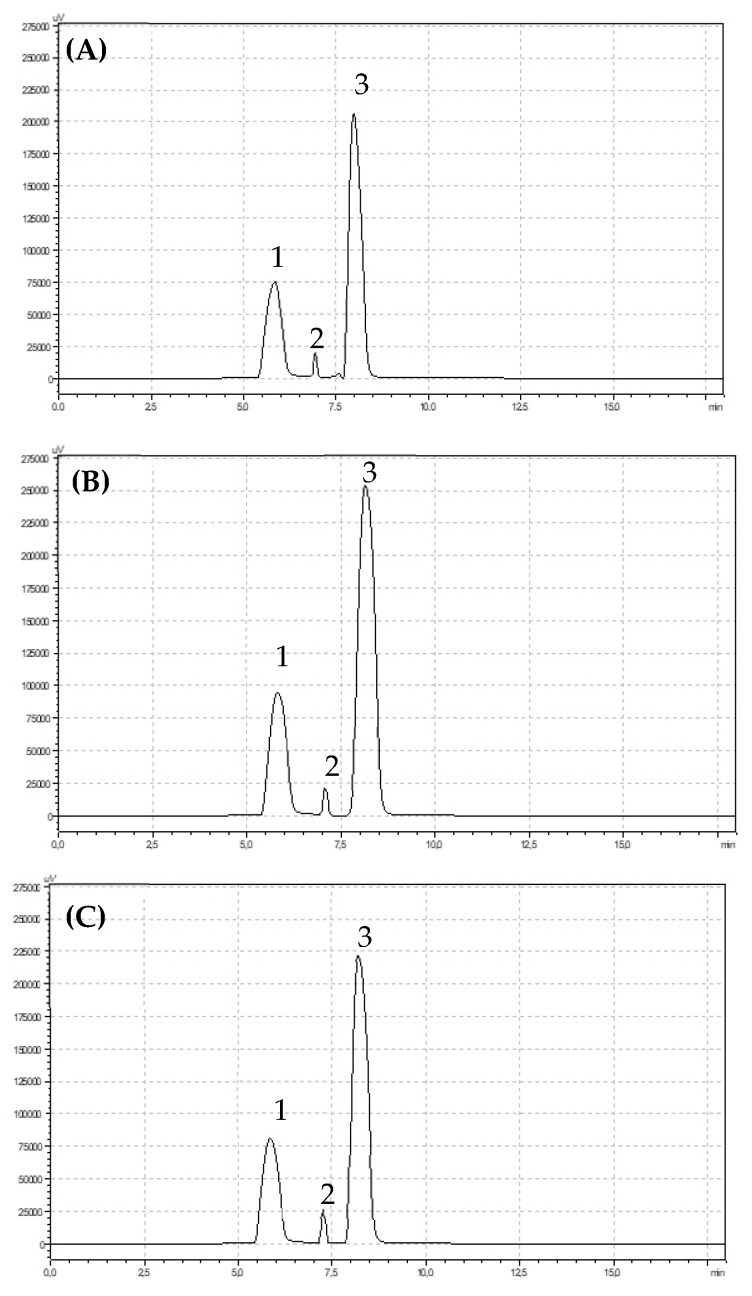
Chromatograms showing retention times for anthocyanins identified in (**A**) organic (**B**) integrated and (**C**) conventional strawberries (1) pelargonidin-3-O-glucoside, (2) pelargonidin-3-O-rutinoside, (3) cyanidin-3-O-glucoside.

**Figure 5 metabolites-10-00102-f005:**
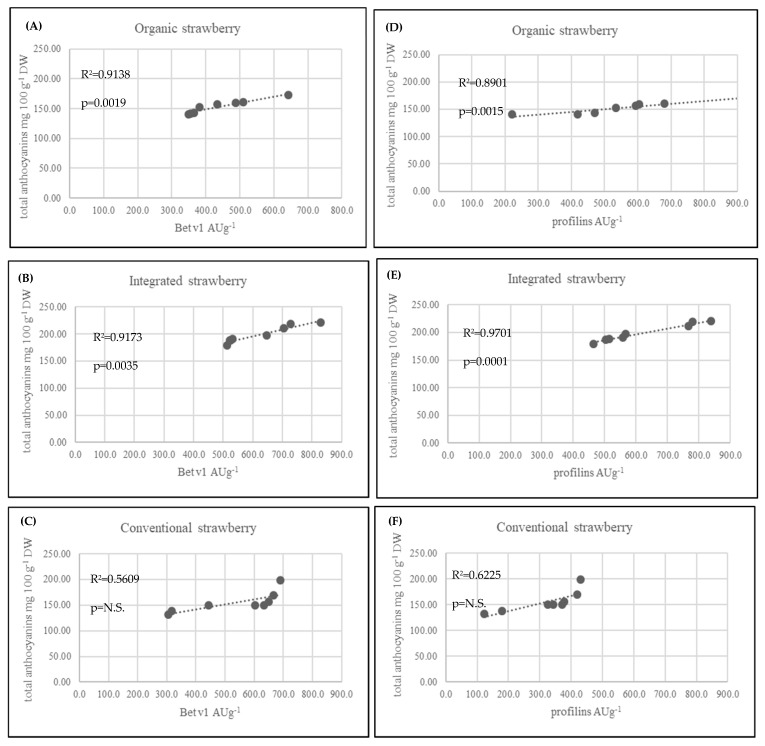
Linear correlation (Pearson’s coefficient R^2^) between allergenic proteins profilins, Bet v 1 and total anthocyanins in strawberry fruits from three cultivation systems.

**Table 1 metabolites-10-00102-t001:** Major agricultural practices in different crop production systems in experimental farms.

Crop Production System	Localization	Soil Type	Forecrop	Irrigation	Organic and Mineral Fertilizers	Plant Protection
**Organic Farms**	Leokadiów (51°24’ N21°48’ E)	Luvisol, pH_(KCl)_ 5.6	Cereals	No	Cow manure 5 t ha^−1^	No
	Ignaców (51°45’ N 20°38’ E)	Luvisol, pH_(KCl)_ 6.1	Fabaceae	No	Cow manure 5 t ha^−1^ Biosol (organic fertiliser) 300 kg ha^−1^	No
	Cyganówka (51°21’ N 21°37’ E)	Luvisol, pH_(KCl)_ 6.2	Potato	No	Cow manure 5 t ha^−1^	No
	Mieczysławów (51°22’ N 21°39’ E)	Luvisol, pH_(KCl)_ 6.3	Fabaceae	No	Cow manure 10 t^−1^ nettle manure 1 t ha^−1^	No
**Conventional Farms**	Dobrosławów (51°22’ N 21°39’ E)	Luvisol, pH_(KCl)_ 5.9	Fabaceae	No	Slurry 25 m^3^ ha^−1^ Hydrocomplex 200 kg ha^−1 ^(N 12%, P_2_O_5_ 11%, K_2_O 18%)	Fungicide 3.9 kg(L) ha^−1^: Vaxiplant SL 1.0 kg ha^−1^ Polyversum WP 0.1 kg ha^−1^; Dymas 2.0 L ha^−1^ Luna Sensation 500 S.C. 0.8 L ha^−1^; Insecticide 2.0 L ha^−1^: Ortus 0.5 S.C. 2.0 L ha^−1^
	Zarzecze (51°24’ N 21°50’ E)	Luvisol, pH_(KCl)_ 6.2	Cereals/Fabaceae	Yes	Slurry 20 m^3^ ha^−1^ Hydrocomplex 200 kg ha^−1^	Fungicide 2.8 kg(L) ha^−1^: Vaxiplant SL 1.0 kg ha^−1^ Luna Sensation 500 S.C. 0.8 L ha^−1^; Scorpion 325 S.C. 1.0 L ha^−1^; Herbicide 2.0 L ha^−1^: Targa Super 0.5 EC 2 L ha^−1^
	Dobrosławów (51°22’ N 21°39’E)	Luvisol, pH_(KCl)_ 6.0	Fabaceae	No	Slurry 35 m^3^ ha^−1^ Hydrocomplex 200 kg ha^−1^	Fungicide 5.8 kg(L) ha^−1^: Vaxiplant SL 1.0 kg ha^−1^ Luna Sensation 500 S.C. 0.8 L ha^−^1; Pomarsol Forte 80 WG 4.0 kg ha^−1^; Insecticide 0.2 kg ha^−1^: Mospilan 20 SP 0.2 kg ha^−1^
	Dobrosławów (51°22’ N 21°39’ E)	Luvisol, pH_(KCl)_ 6.0	Fabaceae	No	Slurry 20 m^3^ ha^−1^ Hydrocomplex 200 kg ha^−1^	Fungicide 1.8 kg(L) ha^−1^: Vaxiplant SL 1.0 kg ha^−1^ Luna Sensation 500 S.C. 0.8 L ha^−1^; Insecticide 0.2 kg ha^−1^: Mospilan 20 SP 0.2 kg ha^−1^; Herbicide 1.0 kg ha^−1^: Agil 100 EC 1.0 kg ha^−1^
**Integrated Farms**	Janów (51°24’ N 21°51’ E)	Luvisol, pH_(KCl)_ 6.2	Cereals	Yes	Manure 25 t ha^−1^ Slurry 20 m^3 ^ha^−1^ Ammonium sulphate 150 kg ha^−1 ^(32% N) Potassium sulfate 150 kg ha^−1^ (50% K_2_O, 45% SO_3_) Superphosphate 150 kg ha^−1 ^(65% P_2_O_5_, 10% CaO)	Fungicide 4.8 kg(L) ha^−1^: Luna Sensation 500 S.C. 0.8 L ha^−1 ^Pomarsol Forte 80 WG 4.0 kg ha^−1^
	Leokadiów (51° 24’ N 21°48’ E)	Luvisol, pH_(KCl)_ 6.2	Potatoes	No	Slurry 20 m^3 ^ha^−1^ Hydrocomplex 200 kg ha^−1^	Fungicide 1.5 L ha^−1^: Teldor 500 S.C. 1.5 L ha^−1^; Insecticide 0.3 kg ha ^1^: Mospilan 0.3 kg ha^−1^
	Polesie (51°24’ N 21°46’E)	Luvisol, pH_(KCl)_ 6.3	Fabaceae	No	Potassium sulfate 150 kg ha^−1^ Hydrocomplex 200 kg ha^−1^	Instecticide 2.3 kg(L) ha^−1^: Ortus 0.5 S.C. 2.0 L ha^−1^ Mospilan 20 SP 0.3 kg ha^−1^;
	Cyganówka (51°21’ N 21°37’ E)	Luvisol, pH_(KCl)_ 6.4	Potatoes	No	Manure 15 t ha^−1^; Slurry 20 m^3 ^ha^−1^ Hydrocomplex 200 kg ha^−1^	Fungicide 0.8 L ha^−1^: Luna Sensation 500 S.C. 0.8 L ha^−1^ Instecticide 0.3 kg ha^−1^: Mospilan 20 SP 0.3 kg ha^−1^

**Table 2 metabolites-10-00102-t002:** The content of dry matter, phenolics compounds, Bet v1 and profilins in experimental strawberry from three cultivation systems.

Compounds/Production System	Organically Cultivated ‘Honeyoe’ (*n*) = 8	Integrated Cultivated ‘Honeyoe’(*n*) = 8	Conventionally Cultivated ‘Honeyoe’(*n*) = 8
dry matter (g 100 g^-1^ FW)	8.71^1 ^± 0.36 a^2^	8.52 ± 0.14 a	8.20 ± 0.27 a
total polyphenols (mg 100 g^−1^ DW)	478.26 ± 11.37 a	499.76 ± 4.59 a	503.49 ± 17.52 a
total phenolic acids	143.76 ± 4.47 b	184.29 ± 6.49 a	178.01 ± 9.01 a
gallic	7.40 ± 1.10 b	8.83 ± 0.65 b	17.80 ± 0.17 a
chlorogenic	72.03 ± 3.69 b	100.12 ± 8.10 a	96.50 ± 6.86 a
*p*-coumaric	23.79 ± 1.48 a	20.24 ± 1.16 ab	19.80 ± 0.74 b
ferulic	19.42 ± 2.41 a	27.00 ± 3.86 a	16.88 ± 3.73 a
ellagic	21.12 ± 1.24 b	28.10 ± 1.83 a	27.03 ± 0.70 a
total flavonoids (mg 100 g^−1^ DW)	334.50 ± 12.24 a	315.47 ± 5.01 a	325.48 ± 11.13 a
quercetin-3-*O*-rutinoside	86.07 ± 5.25 a	59.45 ± 3.67 a	25.53 ± 2.46 a
quercetin-3-*O*-glucoside	29.69 ± 11.05 b	9.09 ± 0.50 b	90.92 ± 2.90 a
kaempferol-3-*O*-glucoside	49.83 ± 3.00 a	35.09 ± 1.67 b	43.14 ± 2.88 ab
myrycetin	0.55 ± 0.03 b	0.43 ± 0.02 c	0.87 ± 0.02 a
luteolin	1.70 ± 0.09 b	1.03 ± 0.20 c	2.85 ± 0.16 a
quercetin	6.07 ± 0.33 a	6.70 ± 1.26 a	2.81 ± 0.08 b
kaempferol	7.21 ± 1.40 a	4.40 ± 0.56 ab	3.84 ± 0.07 b
total anthocyanins (mg 100 g^−1^ DW)	153.38 ± 3.77 b	199.27 ± 5.19 a	155.51 ± 6.92 b
cyanidin-3-*O*-glucoside	119.52 ± 3.13 b	163.27 ± 4.77 a	120.96 ± 6.13 b
pelargonidin-3-*O*-rutinoside	8.62 ± 0.05 ab	8.87 ± 0.10 a	8.47 ± 0.04 b
pelargonidin-3-*O*-glucoside	25.24 ± 0.66 a	27.13 ± 1.71 a	26.07 ± 0.90 a
Bet v1 (ng g^−1^)	376.37 ± 15.79 b	678.28 ± 29.03 a	637.29 ± 42.10 a
profilins (ng g^−1^)	483.49 ± 40.15 a	532.61 ± 34.57 a	417.89 ± 39.15 a

^1^ Data are presented as the mean ± SE with ANOVA *p*-value; ^2^ Means in rows followed by the same letter are not significantly different at the 5% level of probability (*p *< 0.05).

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
