# Peer review of "Evaluation of the Potential Allergenicity of Strawberries in Response to Different Farming Practices"

_metabolites, 2020, doi:10.3390/metabo10030102_

Round 1

Reviewer 1 Report

In the manuscript “Concentration of secondary metabolites, profilins and Bet v1 reflect the allergenic potency of strawberries in vitro” strawberry fruits from the three cultivation systems, organic, conventional and integrated, were tested for their content of biologically active compounds and their allergenic potency. The subject of the work is valuable and very up to date due to the interest in organic food. In my opinion, the article can be accepted for publication after minor revision. My comments on the text are described below:

In my opinion, statistical analysis is sometimes not properly interpreted, especially regarding polyphenolic acids. The most commonly used p value is 0.05 or 0.01. I would not use such divergent p-values (between 0.0001-0.041) to interpret data in one experiment.

I do not think that there is a statistical difference between integrated and conventional fruits for content of total phenolic acids, chlorogenic acid, ellagic acid and total flavonoids, as well as between organic and integrated fruit for gallic acid. For example, for ellagic acid 28.10 ± 1.83 and 27.03 ± 0.7 - Is there statistical difference between them? Really, with such a large standard error; I think that it is not possible.. Especially since for ferrulic acid, 27 ± 3.86 and 16.88 ±  3.37 -  there is a difference. Authors should once again analyze this data in statistical terms.

Additionally, “p” should be in italics.

Indeed, the strawberries from integrated cultivation have the most anthocyanins and have the greatest allergenic potential. But do really these compounds give this effect? Is this hypothesis not too bold? Basically, anthocyanins are known for their anti-inflammatory properties and can be associated with alleviating allergy symptoms. Maybe this is an only coincidence? Analysis of polyphenols in strawberries is just part of the analysis of the ingredients contained in these fruits. Or maybe it's some other additives used in this crop and present in the fruits increase this potential. Especially, that conventional fruits having comparable anthocyanin content as in organic strawberries, they have a much higher level of Bet v1. line 76: authors should complete geographical data for places of cultivation line 97: 2.4 should be Flavonoid content instead of Polyphenol (flavonoid) content, and line 115: 2.5. sholud be Anthocyanin content instead of Polyphenol (anthocyanin) content line 129 H2O instead of H2O some minor errors in bibliography, like line 445: Baran?ski instead BaraÅ„ski, line 455: Mu?oz instead of Muñoz, sometimes dots instead of commas after the journal name, more than one space between words

Author Response

Reviewer no. 1:

Thank you very much for the review and for your positive recommendation to publish our manuscript in the “Metabolites” journal.

Comment 1: “…In my opinion, statistical analysis is sometimes not properly interpreted, especially regarding polyphenolic acids. The most commonly used p value is 0.05 or 0.01. I would not use such divergent p-values (between 0.0001-0.041) to interpret data in one experiment....”

Authors’ response: According to Reviewer suggestion, Authors check a very carefully and repeat all statistical analysis. In table in last column we insert p-value for specific case. Of course in section Material and methods (sub-section 2.7 Statistical analysis) all details about statistical methods are described. The statistical analysis was performed for p=0.05. If the p-value for a given objects (organic, integrated and conventional) was p< 0.05, a specific numeric value was given in tables. If this value was p>0.05, then it was given in Table N.S. After consultation with Statistical consultant Authors decided remove last column from table. Only information below table now is presented and explain about p-value and homogenous groups.

Comment 2: “…I do not think that there is a statistical difference between integrated and conventional fruits for content of total phenolic acids, chlorogenic acid, ellagic acid and total flavonoids, as well as between organic and integrated fruit for gallic acid. For example, for ellagic acid 28.10 ± 1.83 and 27.03 ± 0.7 - Is there statistical difference between them? Really, with such a large standard error; I think that it is not possible.. Especially since for ferrulic acid, 27 ± 3.86 and 16.88 ±  3.37 -  there is a difference. Authors should once again analyze this data in statistical terms…”

Authors’ response: According to Reviewer suggestion all statistical analysis was repeat and carefully checked. All mistakes were corrected. 

Comment 3: “…Additionally, “p” should be in italics.”

Authors’ response: According to Reviewer suggestion p letter in name of phenolic acid: p-coumaric was formatted by italic style.

Comment 4: “…Indeed, the strawberries from integrated cultivation have the most anthocyanins and have the greatest allergenic potential. But do really these compounds give this effect? Is this hypothesis not too bold? Basically, anthocyanins are known for their anti-inflammatory properties and can be associated with alleviating allergy symptoms. Maybe this is an only coincidence? Analysis of polyphenols in strawberries is just part of the analysis of the ingredients contained in these fruits. Or maybe it's some other additives used in this crop and present in the fruits increase this potential. Especially, that conventional fruits having comparable anthocyanin content as in organic strawberries, they have a much higher level of Bet v1…”

Authors’ response: Of course, we agree with Reviewer that anthocyanins are known for their antioxidant, anti-inflammatory and anti-cancer properties. In the presented experiment, we observed the relationship that a higher anthocyanin concentration in strawberry fruit is also manifested in a higher concentration of homologue Bet v1 allergen. Maybe the hypothesis actually is a bit “too bold” that the allergenic properties depend on the anthocyanin concentration. But anthocyanins concentration (increasing or decreasing) could be used as marker for allergenic potency examination of strawberry. Authors change main idea of manuscript into hypothesis as was suggested by Reviewer no. 3.

However, similar relationships in strawberry fruit were observed by Jung Song et al. (2013). They do not clearly indicate a direct relationship between the anthocyanin concentration in fruits and their allergenic effects. On the other hand, in the summary of the work they clearly indicate that changes occurring in ripening strawberry fruits, among others, increased anthocyanin concentration entails metabolic changes in fruit. The effect of these changes is the synthesis of various proteins. It cannot be ruled out that these may be allergenic proteins.

Jun Song, L.L; Kalt, W.; Forney, Ch.; Tsao, R.; Pinto, D.; Chisholm, K.; Campbell, L.; Fillmore, S.; Li, X. Quantitative proteomic investigation employing stable isotope labeling by peptide dimethylation on proteins of strawberry fruit at different ripening stages. Journal of Proteomics, 2013, 94, 219–239.

In another article, Jimenez et al. (2014) show a relationship between anthocyanin compounds and proteins in elderberry fruit. Reducing the concentration of anthocyanins by using of high temperature changed the protein status of elderberry fruit. As the Authors concluded: it could reduce of unpleased effect of elderberry and their compounds. 

Jimenez, P.; Cabrero, P.; Basterrechea, J. E.; Tejero, J.; Cordoba-Diaz, D.; Cordoba-Diaz, M.; & Girbes, T. Effects of short-term heating on total polyphenols, anthocyanins, antioxidant activity and lectins of different parts of dwarf elder (Sambucus ebulus L.). Plant Foods Hum Nutr 2014, 69, 168–174

In next article, Severo et al. (2015) showed a direct relationship between UV radiation and the synthesis of anthocyanin compounds and the production of Fra a1 allergenic protein responsible for the allergic reaction. The authors of the work conclude that this condition is a direct result of environmental stress to which plants are subjected during the growing period.

Severo, J.; de Oliveira, I. R.; Tiecher, A.; Chaves, F. C.; & Rombaldi, C. V. Postharvest UV-C treatment increases bioactive, ester volatile compounds and a putative allergenic protein in strawberry. LWT - Food Science and Technology, 2015, 64 685–692.

Tulipani et al. (2011) indicate a direct relationship between the concentration of anthocyanin compounds and allergenic strawberry fruit in different years. In the experiment, the increase in anthocyanin concentration during strawberry ripening was directly proportional to the increase in the concentration of allergenic proteins and Fra 1 in fruit.

Tulipani, S., Marzban, G., Herndl, A., Laimer, M., Mezzetti, B., & Battino, M. (2011). Influence of environmental and genetic factors on health-related compounds in strawberry. Food Chemistry, 124(3), 906–913.

The new members of the PR-10 family, the Fra a1 proteins, have been identified in strawberry in response to the flavonoid (anthocyanins belong to flavonoids chemicals) biosynthesis pathway, which is essential for the development of color and flavor in fruits

Jain S., Kumar A. The pathogenesis related class 10 proteins in plant defense against biotic and abiotic stresses, Advances in Plants & Agriculture Research, 2015, 3, 1, 1-11.

Comment 5: “…line 76: authors should complete geographical data for places of cultivation line 97:

Authors’ response: As suggested by the Reviewer, all data on geographical information on farms and agro-technical facilities is presented in Table 1 attached to the manuscript.

Comment 6: “…2.4 should be Flavonoid content instead of Polyphenol (flavonoid) content,

Authors’ response:…. As suggested by the Reviewer the sub-section title was changed

Comment 7: “…line 115: 2.5. sholud be Anthocyanin content instead of Polyphenol (anthocyanin) content

Authors’ response:…. As suggested by the Reviewer the sub-section title was changed

Comment 8: “… line 129 H2O instead of H2O

Authors’ response:…. As suggested by the Reviewer the subscript in H2O is formatted.

Comment 9: “…some minor errors in bibliography, like line 445: Baran?ski instead BaraÅ„ski, line 455: Mu?oz instead of Muñoz, sometimes dots instead of commas after the journal name, more than one space between words

Authors’ response:…. As suggested by the Reviewer all minor errors regarding author names, dots and spaces in section’s bibliography have been corrected

Reviewer 2 Report

The manuscript by Aninowski et al. focuses on the comparison of allergen and flavonoid content in strawberries that are produced in different farming settings. The manuscript provides some new interesting data. However, it cannot be accepted in the current form as the authors are confusing Bet v 1 (birch allergen) with Fra a 1 (allergen from strawberry). Both Bet v 1 and Fra a 1 belong to the same family of proteins (PR-10), however Bet v 1 is not present in strawberries. …unless the authors generated genetically modified strawberries that have Bet v 1 incorporated. The manuscript has to be rewritten to make a clear distinction between Bet v 1 and Fra a 1. The authors should follow the nomenclature naming of allergens as outlined at allergen.org.

It also seems that the experimental design is completely wrong in the part related to analysis of Fra a 1 (Bet v 1 ???) content. The authors are using antibodies against Bet v 1, while they should use antibodies against Fra a 1.

It is also not clear whether the antibodies against profilin are directed against Bet v 2 or Fra a 4?

Author Response

Reviewer no. 2

Thank you very much for the review and for your positive recommendation to publish our manuscript in the “Metabolites” journal.

Comment 1: “…The authors are confusing Bet v 1 (birch allergen) with Fra a 1 (allergen from strawberry). Both Bet v 1 and Fra a 1 belong to the same family of proteins (PR-10), however Bet v 1 is not present in strawberries. …unless the authors generated genetically modified strawberries that have Bet v 1 incorporated…”

Authors’ response: The authors agree with the Reviewer opinion. Of course Bet v1 is not the same as specific strawberry allergen Fra a1. In recent years, several cross-reactive allergens have been identified and sequenced in fruits of the Rosaceae family. The major allergens in strawberry (Fra a1) are structural homologs to the birch pollen major allergen Bet v1, which belongs to class 10 of pathogenesis-related proteins. One of the most important strawberry allergen proteins identified as Fra a1 is the first specific allergen of the strawberry to have been identified and characterized. It is well known as homologue of Bet v1 (the major allergen of birch), and therefore, as the latter, is also a PR-10. Because authors performed experiment with one strawberry cultivars (Honeoye) in organic, integrated and conventional farm management, so using of GMO strawberry, especially in organic system is forbidden.

Comment 2: “…The manuscript has to be rewritten to make a clear distinction between Bet v 1 and Fra a 1. The authors should follow the nomenclature naming of allergens as outlined at allergen.org…”

Authors’ response: As author explain above Bet v1 (birch pollen allergen) is used as factor of potential allergy because is well known as homologue Fra a1. In many publication this practice is presented. In the section Materials and method authors underline, that only potential allergenicity is measured in strawberry samples. According to Reviewer suggestion authors replace not correct used word “allergen Bet v1 occurring in strawberry” into “homologue Bet v1” in the whole manuscript.

The following are examples of articles that also used the measurements of the potential allergenicity of the Fra a1 homolog, Bet v1:

Besbes, F., Franz-Oberdorf, K., Schwab W. Phosphorylation-dependent ribonuclease activity of Fra a 1 proteins, Journal of Plant Physiology,  233,  2019, 1-11.

Franz-Oberdorf, K., Eberlein, B., Edelmann, K., Hücherig, S., Besbes, F., Darsow, U., Ring, J., Schwab, W.  Fra a 1.02 Is the Most Potent Isoform of the Bet v 1-like Allergen in Strawberry Fruit. Journal of Agricultural and Food Chemistry, 2016, 64, 18, 3688–3696.

Hyun, T.K.; Kim, J-S. Genomic identification of putative allergen genes in woodland strawberry (Fragaria vesca) and mandarin orange (Citrus clementina), Plant Omics 2011, 4, 428-434.

Musidlowska-Persson, A., Alm, R., & Emanuelsson, C. Cloning and sequencing of the Bet v 1-homologous allergen Fra a 1 in strawberry (Fragaria ananassa) shows the presence of an intron and little variability in amino acid sequence. Molecular Immunology, 2007, 44, 6, 1245–1252.

Comment 3: “…It also seems that the experimental design is completely wrong in the part related to analysis of Fra a 1 (Bet v 1 ???) content. The authors are using antibodies against Bet v 1, while they should use antibodies against Fra a 1…”

Authors’ response: As explained above, Bet v1 is considered a Fra a 1 homologue. In presented manuscript only Bet v1 was measures as the factor of potential allergenic status of strawberries. This is explained in  2.6 section of M&M: Allergenic potency analysis.

Comment 4: “…It is also not clear whether the antibodies against profilin are directed against Bet v 2 or Fra a 4?...”

Authors’ response: the profilins used in this experiment are directed against overall allergenic potency. In the literature they are described as non-specific plant profilins (plant proteins).

Reviewer 3 Report

The paper is mostly well written and relevant. It evaluates the content of secondary metabolites and some proteins related to fruit quality and allergenic potency in strawberry cultivated under three different farming systems.

Nevertheless, I have some serious concerns about the reproducibility of the research, given the scarce details provided about these three farming systems. Authors only refer to them as "Conventional", "Integrated" and "Organic" and very little description is presented. Given that secondary metabolite contents are so dependent on environmental conditions and management practices (which is, in fact, the central point of the manuscript), a detailed description of these crop management practices should be added to the manuscript, including fertilization types and regimes, type of soil, irrigation, pest management, arrange of the plants, amongst many other variables.

On the other hand, the PCA analysis is not adequately performed. Authors use only three points for the plot (maybe the vectors composed of the average of each measured variable) when they should use all the individual replicates and see how they cluster. This is probably the reason why they got 100% of the explained variance in the first 2 components.

In lines 330-334 and later in the manuscript the authors describe a linear relationship (not regression, as stated in line 330) between anthocyanin content and profilins and Bet v1 allergens (proteins). Given the fact that correlation does not imply causation, do the authors have a hypothesis for this correlation and why this correlation is not observed in the conventional system? As this link is especially highlighted, it would be interesting to understand the reasons.

Regarding the title of the manuscript, I think it is not reflecting the content of the manuscript. The term “in vitro” is not applicable as the strawberries were cultivated in field conditions.

Finally, there are some not supported or wrongly written statements in the text:
Integrated/organic/conventional strawberries: bad wording, as the adjectives are applicable to the farming practices and not to the fruits.
The meaning of the term “Integrated” is not defined until line 278-279.
Line 56. Replace “seven proteins” with “seven allergenic proteins” or similar.
Lin4 69. I think it is better to express this idea in terms of hypothesis.
Line 277. What do the authors mean with “pure”?

Author Response

Reviewer no. 3

Thank you very much for the review and for your positive recommendation to publish our manuscript in the “Metabolites” journal.

Comment 1: “…I have some serious concerns about the reproducibility of the research, given the scarce details provided about these three farming systems. Authors only refer to them as "Conventional", "Integrated" and "Organic" and very little description is presented. Given that secondary metabolite contents are so dependent on environmental conditions and management practices (which is, in fact, the central point of the manuscript), a detailed description of these crop management practices should be added to the manuscript, including fertilization types and regimes, type of soil, irrigation, pest management, arrange of the plants, amongst many other variables…”

Authors’ response: Authors agree with Reviewer opinion. Much more information about characteristic of three farm management now is presented in Introduction section, as well all details of cultivation are presented in Table 1: localization of experimental farms, fertilization and protection regime, type of soil, kind of fertilizer used in time of strawberry cultivation.

Comment 2: “…On the other hand, the PCA analysis is not adequately performed. Authors use only three points for the plot (maybe the vectors composed of the average of each measured variable) when they should use all the individual replicates and see how they cluster. This is probably the reason why they got 100% of the explained variance in the first 2 components..…”

Authors’ response: According to Reviewer suggestion the PCA analysis was repeated. The authors used individual repetitions (each 2 points represented one experimental farm where strawberries were cultivated).

Comment 3: “…In lines 330-334 and later in the manuscript the authors describe a linear relationship (not regression, as stated in line 330) between anthocyanin content and profilins and Bet v1 allergens (proteins). Given the fact that correlation does not imply causation, do the authors have a hypothesis for this correlation and why this correlation is not observed in the conventional system? As this link is especially highlighted, it would be interesting to understand the reasons…”

Authors’ response: Authors agree with Reviewer comment. The analysis concerned the correlation between anthocyanin concentration and allergenic factors: Bet v1 and profilin. With all due respect for the Reviewer, the conventional strawberry shows as well correlation (lines 329-331), however, as the authors emphasize in the discussion, it is weak relationship and not statistically significant. To explain such situation more experiment in the future is required. Long term experiment are needed to observe such trends or stabile effects.

Comment 4: “…Regarding the title of the manuscript, I think it is not reflecting the content of the manuscript. The term “in vitro” is not applicable as the strawberries were cultivated in field conditions.

Authors’ response: According to Reviewer suggestion the term “in vitro” was removed from the manuscript title.

Comment 5: “…Finally, there are some not supported or wrongly written statements in the text: Integrated/organic/conventional strawberries: bad wording, as the adjectives are applicable to the farming practices and not to the fruits

Authors’ response: Authors agree with Reviewer comment. On the other hand using of the terms “conventional strawberries”, “organic strawberries” and “integrated strawberries” is term simplification. However, the Authors would like to emphasize that, this is a common practice used in almost all scientific publications. Of course, the term organic / integrated / conventional refers primarily to production systems. However, to avoid constant repetition and awkwardness, e.g. “strawberries from the organic production” are replaced by short term “organic strawberries”. It is similar with other agricultural production systems (integrated and conventional) and their agricultural produce. However, as suggested by the Reviewer, the Authors decided to change everything that is possible in the manuscript text from a simplification of “organic strawberry” into “organically produced strawberry” or “strawberry from organic farming”

Examples:

Crecente-Campo, J., Nunes-Damaceno, M., Romero-Rodríguez, M. A., & Vázquez-Odériz, M. L. (2012). Color, anthocyanin pigment, ascorbic acid and total phenolic compound determination in organic versus conventional strawberries (Fragaria×ananassa Duch, cv Selva). Journal of Food Composition and Analysis, 28(1), 23–30.

Fernandes, V. C., Domingues, V. F., de Freitas, V., Delerue-Matos, C., & Mateus, N. (2012). Strawberries from integrated pest management and organic farming: Phenolic composition and antioxidant properties. Food Chemistry, 134(4), 1926–1931.

Rhainds, M., Kovach, J., & English-Loeb, G. (2002). Impact of Strawberry Cultivar and Incidence of Pests on Yield and Profitability of Strawberries under Conventional and Organic Management Systems. Biological Agriculture & Horticulture, 19(4), 333–353.

Swezey, S. L., Nieto, D. J., & Bryer, J. A. (2007). Control of Western Tarnished Plant Bug Lygus hesperus Knight (Hemiptera: Miridae) in California Organic Strawberries Using Alfalfa Trap Crops and Tractor-Mounted Vacuums. Environmental Entomology, 36(6), 1457–1465.

Cardoso, P. C., Tomazini, A. P. B., Stringheta, P. C., Ribeiro, S. M. R., & Pinheiro-Sant’Ana, H. M. (2011). Vitamin C and carotenoids in organic and conventional fruits grown in Brazil. Food Chemistry, 126(2), 411–416.

Comment 6: “…The meaning of the term “Integrated” is not defined until line 278-279.”

Authors’ response: As suggested by the Reviewer in the Introduction section more information and proper definition of both Integrated and Organic system were added.

Comment 7: “…Line 56. Replace “seven proteins” with “seven allergenic proteins” or similar.

Authors’ response: As suggested by the Reviewer, a correction was made

Comment 8: “…Lin4 69. I think it is better to express this idea in terms of hypothesis

Authors’ response: As suggested by the Reviewer, Authors change main idea of manuscript into hypothesis.

Comment 9: “…Line 277. What do the authors mean with “pure”?

Authors’ response: The word “pure” means more safe for consumers. The authors change the word “pure” into “more controlled”

Round 2

Reviewer 2 Report

The authors answers are not satisfactory. There is still  confusion and lack of clarity with allergen naming. Fra a 1, which is Bet v 1 homolog, is the focus of the manuscript. The authors should not use Fra a 1 and Bet v 1 names interchangeable.

It is not clarified whether the authors used an antibody against Fra a 1 or Bet v 1. ...or it was a cross-reactive antibody that recognizes both Fra a 1 and Bet v 1.

The authors are still incorrectly naming  allergens as "Fra a1" and "Bet v1"

The term "overall allergenic potency" is not clear.

Author Response

Thank you very much for the review and for your positive recommendation to change our manuscript to be ready for publish in the “Metabolites” journal.

Comment 1: “The authors answers are not satisfactory. There is still  confusion and lack of clarity with allergen naming. Fra a 1, which is Bet v 1 homolog, is the focus of the manuscript. The authors should not use Fra a 1 and Bet v 1 names interchangeable.”

Authors’ response: According to the Reviewer's suggestion, the authors removed from the entire text of the manuscript the misleading name of allergens. Now, the name allergens Fra a1 and Bet v1 are not interchangeably used. The subject of the study was Bet v1 not Fra a1.

Comment 2: “It is not clarified whether the authors used an antibody against Fra a 1 or Bet v 1. ...or it was a cross-reactive antibody that recognizes both Fra a 1 and Bet v 1.”

Authors’ response: The authors explain that only Bet v1 was used in the whole experiment. Detailed information is provided in section 2.6. Potential allergenicity analysis, line 225-240 and : “…The following reagents were used: mouse antibodies against Bet v1. In presented manuscript only determination of potentially allergenic protein homologous to Fra a 1 by means of cross-reaction with antibodies against Bet v 1 have been done.”

Comment 3: “The authors are still incorrectly naming  allergens as "Fra a1" and "Bet v1"”

Authors’ response: According to the Reviewer's suggestion, the names of allergens are ordered. Because only  Bet v1 was the subject of the study, this allergen name was retained throughout the manuscript.

Comment 4: “The term “overall allergenic potency” is not clear.”

Authors’ response: According to the Reviewer's suggestion not quite clear term “overall allergenic potency” was replaced by “potential allergenicity”

Reviewer 3 Report

I think the authors have substantially improved the manuscript. Table 1 is critical for the reproducibility of the research and has sufficient information. Yet, I still have some further suggestions to improve the manuscript.

Title: please consider a title that reflects the experiment performed in the manuscript. Like “Evaluation of the allergenic potency of strawberries in response to farming practices” or something similar.

Lines 18-20. Please present the farming methods accordingly and do not express ideas in absolute terms. Please consider rephrasing: “In fruit and vegetable production, three different farming systems are mainly found: conventional, integrated pest management and organic production. The latter is considered the best in terms of …” or something similar.

Line 34. “According to the rules and law”, please be more specific. Which rules and laws?

Lines 70-71. Please consider removing the amended sentence. This is not a hypothesis, and the manuscript didn’t find the link, just a correlation that is not explained.

Line 201. Please remove the word significant.

Line 202. Reference to figure 4 comes before the reference to figures 2 and 3. Please change the numbering accordingly.

Lines 207-208. Please reword “significantly suggest total dissimilarity” with “This suggests significant dissimilarity in the…” or similar.

Line 209. Please remove “In the quantitative system”.

Line 222. Bad wording, “We did not observe”.

Line 223. Check typo, “From”.

Lines 285-286. Please, don’t use absolute terms. Rephrase with: “Organic agriculture is considered one of the best alternatives for sustainable and good quality food production” or similar.

Line 290. Please change “have focused on the theory” with “have shown that…” or similar. If the references showed these results, it is not a theory.

Line 292. Please rephrase “is almost unknown” with “has not been assessed yet” or similar.

Line 293-294. Please change the reference to the “theory” with “Our results are partially in accordance with previous studies” or similar.

Lines 325-327. Please rephrase both sentences to “The anthocyanin synthesis pathway is regulated primarily at the transcriptional level and is induced by UV light” or similar.

Lines 340-352. About the correlation between proteins and anthocyanins. Even though authors cannot suggest the reasons of this correlation, clear sentences should state that 1) further research will be needed to test if this correlation is reflecting direct relationships of biological processes and 2) why do the authors especially highlight this relationship or which are the consequences of this relationship for the consumers.

Lines 352-354. Please rephrase both sentences to “In addition to natural allergens, the plants produced in conventional systems may contain many residual chemicals (e.g. pesticides or derivatives) that also have toxic and allergenic properties” or similar.

Line 407. Please rephrase “confirm the theory” with “support the hypothesis”.

Author Response

Thank you very much for the review and for your positive recommendation to change our manuscript to be ready for publish in the “Metabolites” journal.

Comment 1: “Title: please consider a title that reflects the experiment performed in the manuscript. Like “Evaluation of the allergenic potency of strawberries in response to farming practices” or something similar”

Authors’ response: According to Reviewer suggestion the title of manuscript was changed.

Comment 2: “Lines 18-20. Please present the farming methods accordingly and do not express ideas in absolute terms. Please consider rephrasing: “In fruit and vegetable production, three different farming systems are mainly found: conventional, integrated pest management and organic production. The latter is considered the best in terms of …””

Authors’ response: According to Reviewer suggestion description of farming methods was changed

Comment 3: “Line 34. “According to the rules and law”, please be more specific. Which rules and laws?”

Authors’ response: It is Council Regulation (EC) No 834/2007 of 28 June 2007 on organic production and labelling of organic products and repealing Regulation (EEC) No 2092/91. An appropriate reference source has been included in the text and the list of references

Comment 4: “Lines 70-71. Please consider removing the amended sentence. This is not a hypothesis, and the manuscript didn’t find the link, just a correlation that is not explained.”

Authors’ response: According to Reviewer suggestion sentence was removing from the manuscript text.

Comment 5: “Line 201. Please remove the word significant.”

Authors’ response: According to Reviewer suggestion word “significant” was removing from the manuscript text.

Comment 6: “Line 202. Reference to figure 4 comes before the reference to figures 2 and 3. Please change the numbering accordingly.”

Authors’ response: According to Reviewer suggestion Figure numbering has been improved. Figures appear in the manuscript text in numerical order.

Comment 7: “Lines 207-208. Please reword “significantly suggest total dissimilarity” with “This suggests significant dissimilarity in the…” or similar.”

Authors’ response: According to Reviewer suggestion sentence was corrected

Comment 8: “Line 209. Please remove “In the quantitative system”.”

Authors’ response: According to Reviewer suggestion sentence was corrected

Comment 9: “Line 222. Bad wording, “We did not observe”.”

Authors’ response: According to Reviewer suggestion sentence was corrected

Comment 10: “Line 223. Check typo, “From”.”

Authors’ response: According to Reviewer suggestion word “form” was corrected into “from”

Comment 11: “Lines 285-286. Please, don’t use absolute terms. Rephrase with: “Organic agriculture is considered one of the best alternatives for sustainable and good quality food production”

Authors’ response: According to Reviewer suggestion sentence was corrected

Comment 12: “Line 290. Please change “have focused on the theory” with “have shown that…” or similar. If the references showed these results, it is not a theory.”

Authors’ response: According to Reviewer suggestion sentence was corrected

Comment 13: “Line 292. Please rephrase “is almost unknown” with “has not been assessed yet” or similar.”

Authors’ response: According to Reviewer suggestion sentence was corrected

Comment 14: “Line 293-294. Please change the reference to the “theory” with “Our results are partially in accordance with previous studies”

Authors’ response: According to Reviewer suggestion sentence was corrected

Comment 15: “Lines 325-327. Please rephrase both sentences to “The anthocyanin synthesis pathway is regulated primarily at the transcriptional level and is induced by UV light”

Authors’ response: According to Reviewer suggestion sentence was corrected

Comment 16: “Lines 340-352. About the correlation between proteins and anthocyanins. Even though authors cannot suggest the reasons of this correlation, clear sentences should state that 1) further research will be needed to test if this correlation is reflecting direct relationships of biological processes and 2) why do the authors especially highlight this relationship or which are the consequences of this relationship for the consumers.”

Authors’ response: As suggested by the Reviewer, information on the need for further research in this direction was added to the manuscript. At the same time, the authors want to clarify that the experiment presented by Hjernoe et al. (2006) was the basis for formulating the research hypothesis presented in the presented manuscript. Article of Hjernoe et al. (2006) was cited in the paper towards an attempt to explain the observed phenomenon of anthocyanin and allergenic protein dependence in strawberry fruit. At the same time, information on the practical importance of this phenomenon for consumers has been added into manuscript text.

Comment 17: “Lines 352-354. Please rephrase both sentences to “In addition to natural allergens, the plants produced in conventional systems may contain many residual chemicals (e.g. pesticides or derivatives) that also have toxic and allergenic properties”

Authors’ response: According to Reviewer suggestion sentence was corrected

Comment 18: “Line 407. Please rephrase “confirm the theory” with “support the hypothesis”.”

Authors’ response: According to Reviewer suggestion sentence was corrected.

Round 3

Reviewer 2 Report

The changes are not satisfactory. Please see www.allergen.org for information on correct use allergen nomenclature.